# Effects of Different Parameter Settings for 3D Data Smoothing and Mesh Simplification on Near Real-Time 3D Reconstruction of High Resolution Bioceramic Bone Void Filling Medical Images

**DOI:** 10.3390/s21237955

**Published:** 2021-11-29

**Authors:** Daniel Jie Yuan Chin, Ahmad Sufril Azlan Mohamed, Khairul Anuar Shariff, Mohd Nadhir Ab Wahab, Kunio Ishikawa

**Affiliations:** 1School of Computer Sciences, Universiti Sains Malaysia, Gelugor 11800, Penang, Malaysia; danielcjy96@student.usm.my (D.J.Y.C.); mohdnadhir@usm.my (M.N.A.W.); 2School of Materials and Mineral Resources Engineering, Universiti Sains Malaysia, Engineering Campus, Nibong Tebal 14300, Penang, Malaysia; biokhairul@usm.my; 3Dental Materials Science and Technology Division, Faculty of Dental Medicine, Airlangga University, JI. Prof. Dr. Moestopo No. 47, Surabaya 60132, East Java, Indonesia; 4Department of Biomaterials, Faculty of Dental Science, Kyushu University, 3-1-1 Maidashi, Higashi-ku, Fukuoka 812-8582, Japan; ishikawa@dent.kyushu-u.ac.jp

**Keywords:** 3D reconstruction, 3D data smoothing, mesh simplification, high resolution micro-CT images

## Abstract

Three-dimensional reconstruction plays a vital role in assisting doctors and surgeons in diagnosing the healing progress of bone defects. Common three-dimensional reconstruction methods include surface and volume rendering. As the focus is on the shape of the bone, this study omits the volume rendering methods. Many improvements have been made to surface rendering methods like Marching Cubes and Marching Tetrahedra, but not many on working towards real-time or near real-time surface rendering for large medical images and studying the effects of different parameter settings for the improvements. Hence, this study attempts near real-time surface rendering for large medical images. Different parameter values are experimented on to study their effect on reconstruction accuracy, reconstruction and rendering time, and the number of vertices and faces. The proposed improvement involving three-dimensional data smoothing with convolution kernel Gaussian size 5 and mesh simplification reduction factor of 0.1 is the best parameter value combination for achieving a good balance between high reconstruction accuracy, low total execution time, and a low number of vertices and faces. It has successfully increased reconstruction accuracy by 0.0235%, decreased the total execution time by 69.81%, and decreased the number of vertices and faces by 86.57% and 86.61%, respectively.

## 1. Introduction

Bone void fillers are used to treat bone voids caused by various reasons. Traditionally, doctors and surgeons examine bone defects implanted with a bone void filler by scanning it through a computed tomography (CT) scanner, which results in a stack of two-dimensional (2D) CT images. However, it is difficult to judge the healing progress of the bone defect as there are noises and artefacts present in the 2D CT bone defect images, which obscure the details of the bone defect. Additionally, it is difficult to visualize the bone defect as a whole with the 2D CT image stack. Therefore, by visualizing the 2D CT image stack as a three-dimensional (3D) view, 3D models can help doctors increase the quality of experience and diagnosis accuracy [1,2]. The process of visualizing the 2D CT image stack as a 3D view is called 3D reconstruction.

With the advancement in 3D technologies, more and more domains are adopting 3D reconstruction technologies. As this study emphasizes 3D reconstruction using micro-CT, images which are medical images, 3D reconstruction in the medical imaging area is studied. For example, the annulus fibrosus (AF) in the intervertebral disc (IVD) is reconstructed, visualized, and tracked from diffusion tensor imaging (DTI) images [3]. Furthermore, 3D models generated through 3D reconstruction are also used for measurement purposes in which the measurements are useful in surgical planning [4,5]. Moreover, measurements obtained from the reconstructed 3D models are also beneficial in virtual simulation for preoperative plans and personalized surgery [6,7,8]. 3D reconstruction algorithms are generally grouped into two categories: surface rendering and volume rendering [9,10].

Both methods have different advantages and disadvantages. For instance, if the main focus of the visualization is on the surface and shape of the bone, then surface rendering is more suitable than volume rendering. If the main focus of the visualization is on the internal structures, then volume rendering is more suitable than surface rendering. Therefore, as the main focus is on visualizing the surface of the bone defect, the study will only be focusing on surface rendering methods.

Surface rendering is a 3D reconstruction method that deposits isosurface comprising triangular patches on the segmented region of interest (ROI) per medical image slice. This results in a 3D model made up of isosurface stacks. Standard surface rendering techniques discussed in recent years are the Marching Cubes algorithm and the Marching Tetrahedra algorithm. The Marching Cubes algorithm is a popular surface rendering method applied in the medical field, mainly because of its implementation simplicity and relatively fast reconstruction speed. Marching Cubes uses a cube as a unit when forming the isosurface. The cube configurations formed during the triangulation step can be generally categorized into 15 unique patterns identified in the original Marching Cubes algorithm [11]. However, ambiguity issues will occur during the triangulation step, which leads to the formation of “holes” on isosurfaces.

Despite this, the original Marching Cubes algorithm is widely applied in recent publications for reconstruction purposes. For instance, it is used in the construction of polyhedral element meshes [12]. It is also used for lumbar intervertebral disk herniation diagnosis [2]. Improvements over the original Marching Cubes algorithm are also made in recent years. For example, Wang et al. [13] proposed the application of Laplacian smoothing to overcome the ambiguity issue by eliminating zero probability during isosurface extraction, edge collapse method using Quadric Error Metric [14], and polygon merging method through polygon normal regression in the Marching Cubes algorithm to improve the display and interaction speed. A variation of the Marching Cubes comprising 33 unique cube configurations, called the Marching Cubes 33, is introduced by Chernyaev [15] to cover the majority of the complex trilinear function of topology cases that leads to an ambiguity issue, which is further extended by Custodio et al. [16] by grouping the cube configurations into either a simple leaves triangulation, tunnel triangulation, or interior point leaves triangulation. Wi et al. [17] proposed another improved Marching Cubes algorithm that applies Laplacian smoothing to overcome the ambiguity issue, the edge collapse method using Quadric Error Metric [14], and Taubin smoothing to improve mesh quality. Masala et al. [18] proposed an improved Marching Cubes algorithm by extending the original 15 cube configurations to 21 cube configurations to cover some of the complex topology cases identified by the authors. Wang et al. [19] also proposed an improved Marching Cubes algorithm by extending the 15 cube configurations to 24 cube configurations, which covers more complex triangulation cases, eliminating more “holes”.

On the other hand, Marching Tetrahedra is a variant of Marching Cubes that uses a tetrahedron as a unit instead of a cube for isosurface [20]. One significant advantage of Marching Tetrahedra over Marching Cubes is that there is no ambiguity issue. However, a considerable number of vertices and faces are generated to represent the isosurface. There are also improvements in the Marching Tetrahedra algorithm in recent years. For example, Lu and Chen [21] used bisection iterations to shorten the time needed to calculate the intersection points and proposed a more straightforward redundant vertices elimination strategy as a post-reconstruction step. Bagley et al. [22] proposed an improved Marching Tetrahedra algorithm by applying a simple bisection technique to discover the edge-cut locations, warped and retetrahedralized. Guo et al. [23] proposed another improved Marching Tetrahedra algorithm which involves removing wrongly connected tetrahedrons by looking into the neighborhood relation among the tetrahedrons in all directions. Finally, Ren et al. [24] proposed a more complex improved Marching Tetrahedra algorithm that uses the second-order tetrahedral element C3D10 as a unit during surface reconstruction.

Although many of the improvements focused on increasing the reconstruction accuracy and reducing the reconstruction time for relatively small image datasets, they are not tested with large image datasets, and not many papers made an in-depth study on the effects of the different parameter settings of the improvement methods towards the reconstruction accuracy, reconstruction time, and optimization of vertices and faces. Hence, this study focuses on three objectives:
To compare Marching Cubes and Marching Tetrahedra in terms of reconstruction accuracy for large image dataset;To optimize the better surface rendering technique towards near real-time rendering and higher reconstruction accuracy, lower reconstruction and rendering time, and a lower number of vertices and faces by experimenting on different parameter value combinations;To study the effects of the improvements’ different parameter values on the reconstruction accuracy, reconstruction time, rendering time, and the number of vertices and faces.


## 2. Materials and Methods

A high-resolution micro-CT image dataset is collected from Dr. Khairul Anuar Shariff from the School of Materials and Mineral Resources Engineering, Universiti Sains Malaysia, Engineering Campus. The image dataset consists of 971 high-resolution micro-CT image slices scanned with Skyscan 1076. Every image slice sits at 1400 dots per inch (DPI) for horizontal and vertical resolution. The images are already sorted by their image slice number; hence, they do not need to be rearranged based on their position. The main reason for choosing this image dataset is that it can be reconstructed as a whole unit without running out of memory.

The overall flowchart is as illustrated in Figure 1. All the implementations are made in the matrix laboratory (MATLAB). Before the reconstruction takes place, the images need pre-processing so that the reconstruction algorithms can support them. Firstly, the ROI is labeled using the Canny edge detector with a threshold value of 0.5, which will also label some of the artefacts. The next pre-processing step is thresholding with a threshold value of 0.3, which removes some artefacts. Lastly, area filtering is applied with only one connected component retrieved to further reduce the artefacts. Because the thresholding and area filtering steps will also remove parts of the ROI, the percentage of ROI retained in the image before being concatenated into 3D volumetric data is kept above 90% for all images.

After the pre-processing step, the reconstruction of 3D volumetric data into 3D models takes place for the comparison study. The Marching Cubes code used in this paper is the implementation by Hammer [25]. A 3D model is generated after the reconstruction process, which is then exported out as a Standard Tessellation Language (STL) file using Sven’s implementation [26]. The Marching Tetrahedra code implemented in this paper uses Hammer’s Marching Cubes implementation as a codebase, vertices and faces orientation, and lookup table changes are also made so that it runs as Marching Tetrahedra instead of as Marching Cubes. The code changes refer to the source code written in C by Bourke [27]. The Marching Tetrahedra replicated in this study is the six tetrahedron variant; hence, after the reconstruction process, each tetrahedron configuration is exported as one individual STL file, totaling six STL files. They are then imported into Blender to combine them into one complete model before exporting it as another single STL file. This results in two 3D models, one from Marching Cubes and another one from Marching Tetrahedra. After successfully importing into Meshlab, the 2D images of the 3D models are captured as black and white images. The captured 2D images are then used to calculate the reconstruction accuracy with the 2D image of the bone defect provided in the dataset as ground truth.

This study uses the structural similarity index (SSIM) and multiscale structural similarity index (MS-SSIM) to evaluate the reconstruction accuracy. Both metrics measure the structural similarity over the 2D images, with the main difference in MS-SSIM being that it measures over different image scales through different image sub-sampling processes with low-pass filters [28]. Values of both metrics in MATLAB implementation are already normalized; therefore, no further action is needed. The higher the SSIM and MS-SSIM values, the higher the reconstruction accuracy. Therefore, the algorithm with higher SSIM and MS-SSIM values is the base for improvement, referred to as algorithm A in this chapter.

The improvement comprises applying two additional steps on top of algorithm A: the 3D data smoothing step and the mesh simplification step. The 3D data smoothing step is applied to the 3D volumetric data before the reconstruction step by algorithm A. This step removes noises in the 3D data, which may improve the 3D reconstruction accuracy and solve the ambiguity issue because noises often lead to wrong triangulations. The 3D data smoothing step applied in this paper is a MATLAB built-in function called *smooth3* [29], which convolves the 3D volumetric data with a convolution kernel at a defined size. In this paper, different convolution kernels, namely box and Gaussian, at different convolution kernel sizes (3, 5, and 11) are tested to investigate the effect of different convolution kernels and their sizes on the reconstruction accuracy. Convolution kernel sizes 3 and 5 are standard kernel sizes. Another commonly used kernel size is 10, but because *smooth3* only accepts odd-numbered convolution kernel size, size 11 is selected.

The mesh simplification step, which is applied after the reconstruction step using algorithm A, is a commonly applied method in reducing the number of vertices and faces whilst retaining the overall shape of the 3D model. This method is useful, especially when the 3D model is 3D printed, significantly reducing the slicing time. The mesh simplification step applied in this paper is also a MATLAB built-in function called *reducepatch* [30]. Unfortunately, there is no proper documentation on this particular MATLAB function, but based on the output of the 3D model, it is speculated that an adaptive remeshing based on a criterion or an error metric is applied. This MATLAB function accepts three parameters: a list of vertices, a list of faces, and a reduction factor between 0.0 and 1.0. The lists of vertices and faces are obtained through the reconstruction step. The reduction factor depicts what percentage of vertices and faces should be left after the reduction step. For example, a reduction factor of 0.1 means only 10% of the vertices and faces are left after the reduction step. In this paper, different reduction factors, starting with 0.1 and ending at 0.9, incrementing by 0.1 for each test case, are experimented on to study the effect of different reduction factors on the reconstruction accuracy.

After the mesh simplification step, the 3D models are exported as STL files and imported into Meshlab to capture the 2D black and white images. These images will be used in calculating the SSIM and MS-SSIM. In this paper, the total number of parameter value combinations tested are 2 different convolution kernels multiplied by 3 different convolution kernel sizes multiplied by 9 reduction factors, which is 54.

In an attempt towards making the reconstruction process for high-resolution medical images near real-time, the whole experiment is made using a laptop with 12 gigabytes (GB) random-access memory (RAM), 6 central processing unit (CPU) cores, and a NVIDIA Geforce RTX2060 graphics processing unit (GPU). MATLAB code changes are also made accordingly to support GPU and parallel processing. Firstly, all the arrays used in the code are defined as *gpuArray* objects stored in the GPU. This allows the code to run in the GPU. It is to be noted that not all functions in MATLAB support *gpuArray* objects and that MATLAB only supports the CUDA-enabled NVIDIA GPU [31]. Secondly, the looping over the image stacks during the pre-processing step is changed from *for* to *parfor* to support parallel processing [32]. The for-loop iterations are executed over a parallel pool of six workers defined by MATLAB during code execution. Lastly, although very minimal code changes were made, the reconstruction function used is the vectorized version of the original algorithm A. MATLAB is optimized to perform operations over array-based objects like matrices and vectors [33]; thus, any loops that can be vectorized will perform much faster than the loop itself. However, only loops that involve basic arithmetic operation over the matrix or vector can be vectorized. The speed-up due to the changes stated above can be evaluated by recording the reconstruction time and rendering time for both the original and improved algorithm A.

The reconstruction time refers to the total time taken from loading the image stack until the reconstruction ends. The reconstruction time does not cover the time to export the 3D models as STL files and is recorded in seconds. It is tabulated as the average over five runs. The rendering time refers to the total time taken to load the 3D model in 3D software, which refers to the time taken to load the 3D model as STL files in Meshlab. It is also recorded in seconds and tabulated as the average over five runs.

In addition to the reconstruction accuracy, the reconstruction time, and rendering time, the number of vertices and faces are also recorded for 3D models reconstructed with original and improved algorithm A. They are recorded by loading the 3D models in Meshlab. Finally, graphs are plotted to study the effect of 3D data smoothing and mesh simplification on the reconstruction accuracy, reconstruction time, rendering time, and the number of vertices and faces.

## 3. Results

For the table results in this chapter, MC refers to Marching Cubes, MT refers to Marching Tetrahedra, S refers to convolution kernel size for *smooth3*, and RF refers to the reduction factor for *reducepatch*.

### 3.1. Marching Cubes vs. Marching Tetrahedra

According to the results in Table 1, the 3D model reconstructed with Marching Cubes has a higher SSIM and MS-SSIM value than the 3D model reconstructed with Marching Tetrahedra, albeit by a minimal margin. This means Marching Cubes has higher reconstruction accuracy than Marching Tetrahedra. Reconstruction time, rendering time, and the number of vertices and faces are not compared here as the reconstruction accuracy is given the highest emphasis in this comparison study. Marching Cubes will be used as a codebase for improvement.

### 3.2. Proposed Improved Marching Cubes with Different Parameter Value Combinations

The reconstructed 3D models with different parameter value combinations are as illustrated in Figure 2. Visually, as there is no difference between smoothing with box convolution kernel and Gaussian convolution kernel, all the images in Figure 2 represent both convolution kernels. Based on the results tabulated in Table A1, two-parameter value combinations are tied for the highest SSIM and MS-SSIM, which are Marching Cubes with a convolution kernel box size 11 and a reduction factor of 0.1, and Marching Cubes with a convolution kernel Gaussian size 11 and a reduction factor of 0.1, with both having a SSIM value of 87.77 and MS-SSIM value of 82.48. Between these two combinations, the one with the lower reconstruction and rendering time is Marching Cubes with a convolution kernel Gaussian size 11 and a reduction factor of 0.1. Regardless of the convolution kernel, as long as the size and reduction factors are the same, they will have the same SSIM, MS-SSIM, and number of vertices and faces.

Amongst all the parameter value combinations, the one with the fastest total execution time (reconstruction time added with rendering time) is Marching Cubes with a convolution kernel box size 3 and a reduction factor of 0.6, with the reconstruction time being 99.41 s and the rendering time being 11.13 s, totaling 110.54 s. As for the parameter value combination with the lowest number of vertices and faces, Marching Cubes with a convolution kernel box size 11 and a reduction factor of 0.1, and Marching Cubes with a convolution kernel Gaussian size 11 and a reduction factor of 0.1 are tied for having 315,490 vertices and 629,003 faces. Between these two, Marching Cubes with a convolution kernel Gaussian size 11 and a reduction factor of 0.1 has a lower total execution time.

To determine which parameter value combination is the most optimal to achieve high reconstruction accuracy, fast reconstruction and rendering time, and a low number of vertices and faces, firstly, the percentages of increase for the SSIM and MS-SSIM values, and percentages of decrease for the total time taken (reconstruction time added with rendering time), and the number of vertices and faces, are calculated for every parameter value combination. After calculating the percentages of increase and decrease, any negative values will be replaced with the value −1, and positive values are divided by 100 to become decimal form. These values are then multiplied with a weightage score. The weightage scores for SSIM and MS-SSIM are 0.3 and 0.3, respectively, totaling 0.6. The weightage score for the total time taken is 0.3, and the weightage scores for the number of vertices and faces are 0.05, respectively, totaling 0.1. Thus, the total weightage score is 1.0. After multiplying with the respective weightage score, the values are added to get the final score for every parameter value combination. The scores are tabulated in Table A2.

Based on the scores in Table A2, Marching Cubes with a convolution kernel Gaussian size 5 and a reduction factor of 0.1 has the highest score, obtaining 0.2962 out of 1.0. This means that this combination is the most optimal in achieving a reconstructed 3D model with high accuracy, low reconstruction and rendering time, and a low number of vertices and faces. As well as this, the proposed improvement (Marching Cubes with a convolution kernel Gaussian size 5 and a reduction factor of 0.1) managed to increase the SSIM and MS-SSIM values by 0.011% and 0.036%, respectively, decrease the total execution time by 69.81%, and decrease the number of vertices and faces by 86.57% and 86.61%, respectively. This means that the number of vertices and faces can be reduced by 86.57% and 86.61%, respectively, and still have relatively high reconstruction accuracy, which means a lot of the vertices and faces in 3D models reconstructed with the original Marching Cubes are unnecessary. They can be safely removed without affecting the shape, boundary, and accuracy of the 3D models. It is noticed that the scores for the convolution kernel box size 3 and Gaussian size 3 are all negative values across all the reduction factors. This happens when either one of the percentages of increase or decrease is a negative value, resulting in its value being replaced with −1 during the calculation of the combination score. This is to penalize the decrease in the reconstruction accuracy. Based on Table A1, it seems that either the SSIM or MS-SSIM or both the metrics’ values are lower than the original Marching Cubes algorithm, therefore, resulting in negative values when calculating the percentage of increase in the SSIM and MS-SSIM values.

Additionally, it is noticed that the scores decrease when the reduction factor increases. This is mainly due to the increase in the total execution time and the number of vertices and faces as the reduction factor increases. This is normal as the larger the reduction factors, the higher the number of vertices and faces retained during the mesh simplification step. It is also noticed that the scores for the convolution kernel size 5 are higher than the scores for the convolution kernel size 11. This is largely due to the drastic increase in total execution time as shown in Table A1.

#### 3.2.1. Effect of Convolution Kernel Size and Reduction Factor on Reconstruction Accuracy

In Figure 3 and Figure 4, the convolution kernel size and reduction factor for both the convolution kernel box and Gaussian are merged into one as both have the same SSIM values. Based on Figure 3, the SSIM values across all reduction factors increased from the convolution kernel size 3 to convolution kernel size 5 before increasing to 87.77% for the convolution kernel size 11, except the reduction factors of 0.2, 0.3, and 0.4 whereby their SSIM values for the convolution kernel size 3 are slightly lower than the SSIM values for the reduction factors of 0.1, 0.5, 0.6, 0.7, 0.8, and 0.9. In Figure 4, the SSIM values across all three convolution kernel sizes remained near-constant as the reduction factor increased, except for the convolution kernel size 3, which showed a slight variant in the SSIM values from the reduction factor of 0.1 to 0.5, before remaining constant from 0.6 to 0.9.

In Figure 5 and Figure 6, the convolution kernel size and reduction factor for both the convolution kernel box and Gaussian are combined as both have the same MS-SSIM values. Based on Figure 5, the MS-SSIM values for reduction factors of 0.1, 0.2, 0.3, and 0.4 showed a slight increase from the convolution kernel size 3 to the convolution kernel size 5, before showing another slight increase from the convolution kernel size 5 to the convolution kernel size 11. This is different for the remaining reduction factors whereby the MS-SSIM values showed a big jump from convolution kernel size 3 to the convolution kernel size 5, before showing the same behavior, which is a slight increase from the convolution kernel size 5 to the convolution kernel size 11. In Figure 6, the MS-SSIM values showed the same behavior as the SSIM values for the convolution kernel sizes 5 and 11, whereby the values remained near-constant as the reduction factor increased, and that only the convolution kernel size 3 showed an extensive variant in the MS-SSIM values when the reduction factor increased from 0.4 to 0.5.

#### 3.2.2. Effect of Convolution Kernel Size and Reduction Factor on Reconstruction Time

Another four graphs are plotted based on the results obtained in Table A1. The first graph is plotted with the reconstruction time against the convolution kernel size for the convolution kernel box. The second graph is plotted with the reconstruction time against the reduction factor also for the convolution kernel box. The third and fourth graphs are essentially the same thing as the first and second graphs but for the convolution kernel Gaussian.

Referring to Figure 7, it seems that, overall, the reconstruction time increased when the convolution kernel box size increased for all reduction factors. All the reduction factors showed a significant increase in the reconstruction time when the convolution kernel box size increased from 5 to 11 compared to when convolution kernel box size increased from 3 to 5, except for the reduction factor of 0.7, whereby it showed a minor increase in the reconstruction time when the convolution kernel size increased from 5 to 11 compared to when the convolution kernel size increased from 3 to 5. Based on Figure 8, for the convolution kernel box size 3, other than the reconstruction time showing an increase as the reduction factor increased from 0.6 to 0.7, the rest showed a decrease in the reconstruction time as the reduction factor increased. For the convolution kernel box size 5, the line graph is much more complicated, with the reconstruction time showing an increase when the reduction factor increased from 0.2 to 0.3, 0.4 to 0.5, 0.6 to 0.7, and 0.8 to 0.9, while the rest showed a decrease in the reconstruction time. For the convolution kernel box size 11, the line graph showed a consistent increase in the reconstruction time when the reduction factor increased from 0.2 to 0.4 and from 0.7 to 0.9, while the rest showed a consistent decrease in the reconstruction time. The position of the line graph for the convolution kernel box size 11 supports the description for Figure 7, whereby it is clearly above the rest of the line graphs.

Based on Figure 9, all the reduction factors showed similar behavior as the convolution kernel box whereby a significant increase in the reconstruction time is seen when the convolution kernel size increased from 5 to 11 compared to when the convolution kernel size increased from 3 to 5, except for two reduction factors. The reduction factor of 0.4 showed a decrease in the reconstruction time when the convolution kernel Gaussian size increased from 3 to 5 before showing an increase in the reconstruction time when the convolution kernel size increased from 5 to 11. The reduction factor of 0.8, on the other hand, showed a near-linear increase in the reconstruction time as the convolution kernel Gaussian size increased. The decrease in the reconstruction time for the reduction factor of 0.4 when the convolution kernel Gaussian size increased from 3 to 5 is seen in Figure 10 whereby the line graph for the convolution kernel size 3 is above the line graph for the convolution kernel size 5 at the reduction factor of 0.4. Referring to Figure 10, besides the same placement of line graph for the convolution kernel Gaussian size 11 as the convolution kernel box size 11 in Figure 9, the line graphs in Figure 10 are also complex. For the convolution kernel Gaussian size 3, the reconstruction time increased when the reduction factor increased from 0.1 to 0.3, from 0.6 to 0.7, and from 0.8 to 0.9, while the rest showed a decrease in the reconstruction time. For the convolution kernel Gaussian size 5, the reconstruction time increased when the reduction factor increased from 0.1 to 0.3, from 0.4 to 0.5, and from 0.7 to 0.8, while the rest showed a decrease in the reconstruction time. The convolution kernel Gaussian size 11, on the other hand, showed an increase in the reconstruction time when the reduction factor increased from 0.1 to 0.2, from 0.3 to 0.5, and from 0.6 to 0.8, while the rest showed a decrease in the reconstruction time.

#### 3.2.3. Effect of Convolution Kernel Size and Reduction Factor on Rendering Time

Different from Figure 7 and Figure 8, Figure 11 and Figure 12 show more direct lines. In Figure 11, all the reduction factors decreased the rendering time as the convolution kernel box size increased. The placement of the lines in the graph is higher as the reduction factor increases, which can be seen more clearly in Figure 12. In Figure 12, for all the convolution kernel box sizes, the rendering time increased logarithmically as the reduction factor increased. The lines show signs of plateauing at the reduction factor of 0.6 before starting to plateau at the reduction factor of 0.7 onwards, except for the convolution kernel box size 3 where it showed a slight variant in the rendering time from the reduction factors of 0.7 to 0.9. Moreover, there is a slight drop in the rendering time for the convolution kernel box sizes 5 and 11 when the reduction factor increased from 0.8 to 0.9.

Figure 13 and Figure 14, show nearly the same line shapes like the ones in Figure 11 and Figure 12, respectively. In Figure 13, the rendering time decreases as the convolution kernel Gaussian size increases for all the reduction factors. Similarly, the placement of the lines in the graph shows that the rendering time increases as the reduction factor increases, which can be seen in Figure 14. Figure 14 also shows a logarithmic increase in the rendering time as the reduction factor increases for all the convolution kernel Gaussian sizes. However, for the convolution kernel Gaussian, only kernel size 11 slightly decreased the rendering time when the reduction factor increased from 0.8 to 0.9.

#### 3.2.4. Effect of Convolution Kernel Size and Reduction Factor on Number of Vertices and Faces

Figure 15 and Figure 16 showed very similar line shapes like the ones in Figure 11 and Figure 13, and Figure 12 and Figure 14, respectively. Based on Figure 15, all the reduction factors showed a decrease in vertices as the convolution kernel size increases for both box and Gaussian since both convolution kernels have the same number of vertices. The placement of the lines also showed that the number of vertices increases as the reduction factor increases, which is expected as more vertices are retained when the reduction factor increases. In Figure 16, the increase in the number of vertices as the reduction factor increases can be seen, in which the increase is logarithmic. The reduction factor of 0.6 is where the lines showed signs of plateauing and completely plateaued from the reduction factor of 0.7 onwards.

Figure 17 and Figure 18 show the same line patterns as in Figure 15 and Figure 16, respectively. This is because faces are dependent on the vertices, hence, exhibiting the same pattern. In Figure 17, the number of faces decreases as the convolution kernel size increases for all reduction factors. Similarly, in Figure 18, the number of faces showed a logarithmic increase as the reduction factor increases for all the convolution kernel sizes in both the convolution kernel box and Gaussian. The line plateaued at a reduction factor of 0.7 onwards.

### 3.3. Marching Cubes vs. Wang et al. vs. Wi et al. vs. Proposed Enhancement

As well as comparing the original Marching Cubes and the proposed enhancement, works by Wang et al. [13] and Wi et al. [17] are replicated and compared in this study. Their works were selected to replicate as both involve the smoothing step and mesh decimation step in their improvement, and the proposed enhancement also involves smoothing and mesh decimation; hence, a direct comparison can be made. The result of the comparison is illustrated in Figure 19, Figure 20 and Figure 21. The surfaces of the reconstructed 3D models are illustrated in Figure 22.

Based on Figure 19, the proposed enhancement (Marching Cubes with 3D data smoothing of Gaussian convolution kernel size 5 and mesh simplification with reduction factor 0.1) has higher SSIM and MS-SSIM values than original Marching Cubes, improvement by Wang et al. [13], and improvement by Wi et al. [17], albeit the difference is minimal (less than 1.0%). In Figure 20 and Figure 21, the proposed enhancement has a lower reconstruction time, rendering time, and the number of vertices and faces compared to the original Marching Cubes, improvement by Wang et al. [13], and improvement by Wi et al. [17]. It is to be noted that the replication of methods proposed by Wang et al. and Wi et al. are made in MATLAB, and different evaluation metrics are applied; hence, there may be a difference in terms of implementation, which explains why their implementation has a lower SSIM and MS-SSIM.

## 4. Discussion

When progressing towards near real-time 3D reconstruction for large, high-resolution medical images, many methods can be employed to speed up the reconstruction process. Two methods are applied in this paper, one is GPU processing, another is vectorization. GPU processing is a common technique applied to speed up the reconstruction process which heavily relies on the GPU. Parallel processing is one of the many approaches in GPU processing. In MATLAB, parallel looping relies heavily on the number of cores supported by the CPU. The higher the number of CPU cores, the higher the number of workers in the MATLAB parallel pool, hence, the faster the processing is. As for vectorization, this is mainly effective for MATLAB implementations. This results in a great reduction in total execution time. However, the total execution time can be further decreased by exploring other methods, like downsizing or downsampling the medical images before the reconstruction process takes place. CPU with more cores can also be used to increase the number of workers in the MATLAB parallel pool. A stronger GPU with more memory can also be tested on.

Based on Figure 3, two possible explanations can be made on the shape of the lines: One, the increase in the SSIM values is proportional to the increase in the convolution kernel size. This suggests that if the convolution kernel size is increased further, the SSIM value growth will follow an exponential growth; Two, the graph does not exhibit exponential growth, meaning to say that the SSIM values may not increase exponentially as the convolution kernel size increases. This is largely due to the lack of results as only three convolution kernel sizes are tested. Additionally, because there is a huge jump in kernel size from 5 to 11, and that is exactly where the signs of exponential growth started showing up as illustrated in the figure, it is difficult to judge the growth rate of SSIM when the convolution kernel size increases. Nevertheless, it is safe to say that the larger the convolution kernel size, regardless of whether it is a box or Gaussian, the higher the SSIM values, which means the higher the reconstruction accuracy. A possible explanation for this is that as the convolution kernel size increases, more areas are covered by the convolution kernel during the smoothing step, hence, having a higher tendency to remove noises in the 3D volumetric data.

Figure 5, however, is a different story from Figure 3, whereby the lines showed signs of logarithmic growth. Even so, not all reduction factors exhibit logarithmic signs. Again, it is difficult to judge the growth of MS-SSIM with just three convolution kernel sizes. But it is safe to say that the larger the convolution kernel size, the higher the MS-SSIM values are. One possible explanation for this is the same as the explanation for Figure 3.

As for the lines in Figure 4 and Figure 6, it is clear that the increase in the reduction factor does not equate to an increase in the SSIM and MS-SSIM values. This means the reduction factor for the mesh simplification step has a near-to-no influence on the reconstruction accuracy. It is safe to say that this statement stays true for all convolution kernel sizes as only the convolution kernel size 3 showed a slight variation in values whereas the convolution kernel sizes larger than that remained constant across all nine reduction factors. A possible explanation for this is that mesh simplification methods work in such a way that they preserve the shape and boundary of the 3D models as much as possible whilst reducing the vertices and faces, hence, resulting in nearly the same 3D model shape. Moreover, as the comparison is made between two 2D images, it does not compare well for simplified mesh models.

For Figure 7 and Figure 9, the same logic and explanation can be applied here from Figure 3: One, it is difficult to tell whether the growth is exponential or otherwise as only three convolution kernel box sizes for both box and Gaussian are tested; Two, it is safe to say that the larger the convolution kernel size for both box and Gaussian, the longer the reconstruction time. One possible explanation for this is that more calculations are involved when performing convolution over the 3D volumetric data using a larger convolution kernel size, therefore, more time is needed to complete the convolution process.

In Figure 8, because the lines do not exhibit a specific pattern in shape, it is difficult to judge the effect of the reduction factor on the reconstruction time. However, the reconstruction time for reduction factors above 0.5 are relatively lower than the ones below 0.5, hence one possible conclusion can be drafted here whereby the time needed to complete the reconstruction process favors higher reduction factors. One possible explanation for this is that as the reduction factor increases, the faster the mesh simplification process ends as it will be able to reach the target number of vertices and faces earlier. However, this explanation only holds true for the convolution kernel box. Referring to Figure 10, the reconstruction times for reduction factors above and below 0.5 showed no signs of being relatively higher or lower than the other half. There is nearly no similarity between all three lines, except for reduction factors 0.5 to 0.6 whereby all three lines dropped, albeit at a different rate. Hence, one possible conclusion that can be drawn from this is that the “right” reduction factor for faster reconstruction time when convolution kernel Gaussian is used is 0.6.

Both graphs in Figure 11 and Figure 13 showed similarity in terms of line shape and pattern whereby the larger the convolution kernel size for both the convolution kernel box and Gaussian, the lower the rendering time for all reduction factors. As the rendering time is closely related to the size of the 3D model, which is equivalent to the number of vertices and faces, one possible explanation on this will need to be explained together with graphs in Figure 15 and Figure 17. In Figure 15 and Figure 17, the same explanation can be applied whereby the larger the convolution kernel size for both box and Gaussian, the lower the number of vertices and faces. The lower the number of vertices and faces, the smaller the 3D model sizes are, which means the faster the rendering process of the 3D models. The reason behind this is that as the convolution kernel size increases, more areas are covered by the convolution kernel, therefore, resulting in more noise being removed during the smoothing step. When more noises are removed from the 3D volumetric data, after the reconstruction process, lesser vertices and faces are needed to represent the 3D models as the removed noises are not reconstructed.

Similarly, both graphs in Figure 12 and Figure 14 showed similarity in terms of line shape and pattern, which is also the same case for graphs in Figure 16 and Figure 18. All four graphs showed that the rendering time and the number of vertices and faces increase as the reduction factor increases for both convolution kernels. This is because as the reduction factor increases, the number of vertices and faces retained increases too, which leads to an increase in the rendering time as the 3D models’ sizes increase as well. However, this is only up until the reduction factor of 0.7 whereby the increase plateaued as it is a logarithmic growth. One possible explanation for this is that when the reduction factor decreases from 1.0, in which the number of vertices and faces are not reduced at all, to 0.9, 0.8, and then to 0.7, the further decrease in the number of vertices and faces do not conform to the criterion or the error metric that preserves the shape and boundary of the 3D model; thus, the number of vertices and faces remained the same from the reduction factors of 0.7 to 0.9. When the number of vertices and faces remained constant, the size of the 3D models remained constant. When the size remained constant, technically the rendering time for those 3D models should remain constant as well. In this case, there are slight variations in the rendering time across the reduction factors of 0.7 to 0.9. This is because the recorded rendering time changes per run, even if the tabulated rendering time is an average over five runs, if one of the runs is recorded to have a slightly higher time than the usual recorded time range, it will easily affect the averaged time.

In previous discussions, it is mentioned that the SSIM and MS-SSIM values, and the number of vertices and faces, are the same even though different convolution kernels are used across the same convolution kernel size. This remains true in both quantitative analysis and qualitative analysis. There is no difference between either 3D models when observed with human eyes. When the list of vertices and faces are observed for both 3D models, they are of the same values. This means that in the context of 3D data smoothing, specifically in MATLAB’s implementation, a different convolution kernel does not have any effect on the reconstruction accuracy, but it affects the reconstruction time.

For future studies, several things can be considered:
Image downsampling or downsizing before reconstruction;Stronger GPU and CPU with more cores;Experimenting with more convolution kernel sizes;Different mesh simplification approaches;Testing with more medical image datasets that can be processed at the same time;Implementing with languages like C++;Different metrics to evaluate the reconstruction accuracy;Different code optimization approaches;Support different GPU products like GPU from AMD.


## 5. Conclusions

In conclusion, Marching Cubes have higher reconstruction accuracy than Marching Tetrahedra for large bone defect medical images. The proposed improvement, which involves 3D data smoothing with convolution kernel Gaussian size 5 and mesh simplification with a reduction factor of 0.1, is the most optimal parameter value combination in achieving a balance between high reconstruction accuracy, low reconstruction and rendering time, and a low number of vertices and faces. With help from GPU, parallel processing, and vectorization, the proposed enhancement has the potential in the near real-time reconstruction of large bone defect medical images. 

Based on the obtained results, it can also be concluded that:
The larger the convolution kernel size, the higher the reconstruction accuracy;The reduction factor does not affect the reconstruction accuracy;The larger the convolution kernel size, the higher the reconstruction time;The reduction factor has an effect on the reconstruction time but no specific growth pattern can be deduced from the graphs; thus, the effect is random;The larger the convolution kernel size, the lower the rendering time;The higher the reduction factor, the higher the rendering time, up until the reduction factor of 0.7 where it stopped increasing;The larger the convolution kernel size, the lower the number of vertices and faces;The higher the reduction factor, the higher the number of vertices and faces, up until the reduction factor of 0.7 where it stopped increasing;Different convolution kernels do not affect the result of the reconstruction qualitatively and quantitatively, except for the reconstruction time.

## Figures and Tables

**Figure 1 sensors-21-07955-f001:**
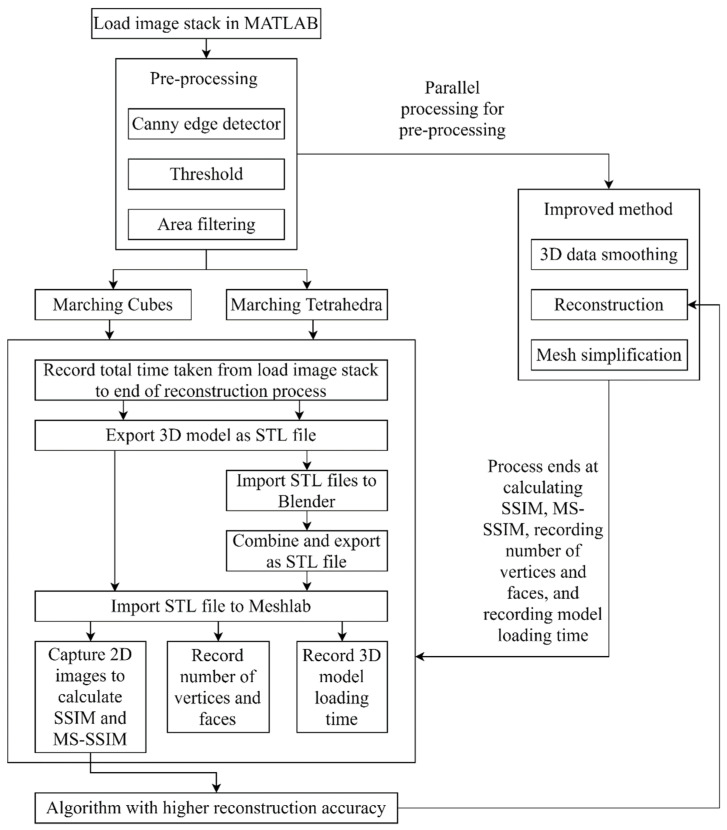
Overall flowchart of methodology.

**Figure 2 sensors-21-07955-f002:**
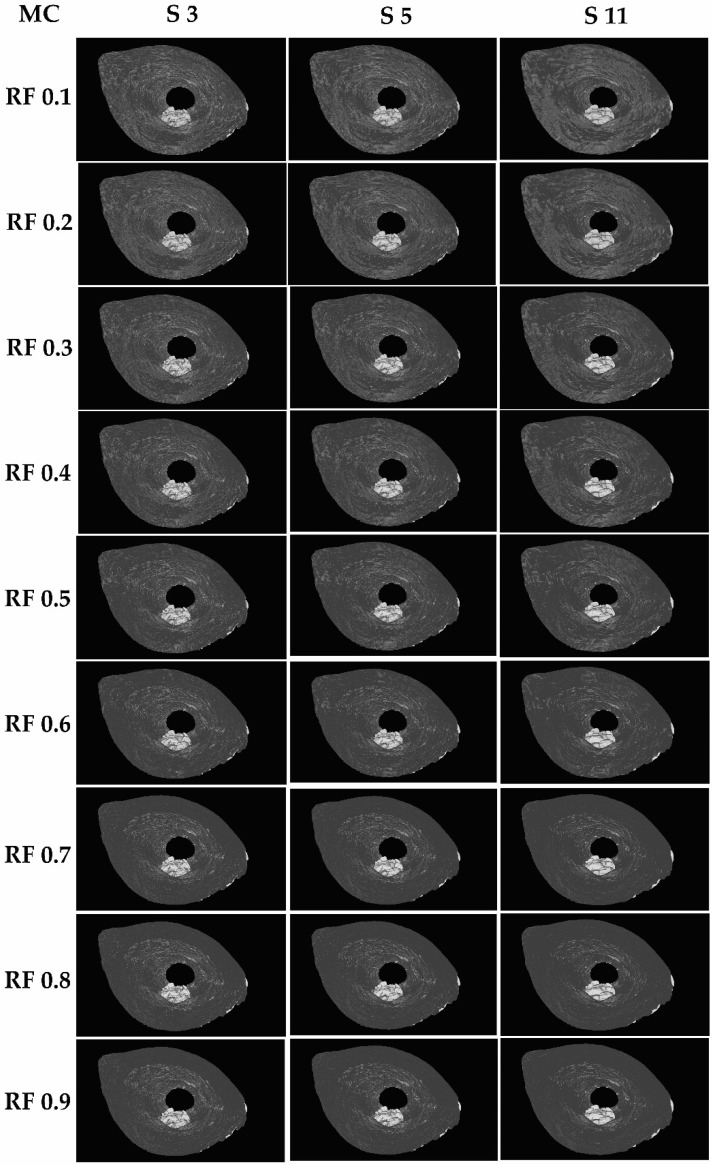
3D models reconstructed with proposed improvement of different parameter value combinations.

**Figure 3 sensors-21-07955-f003:**
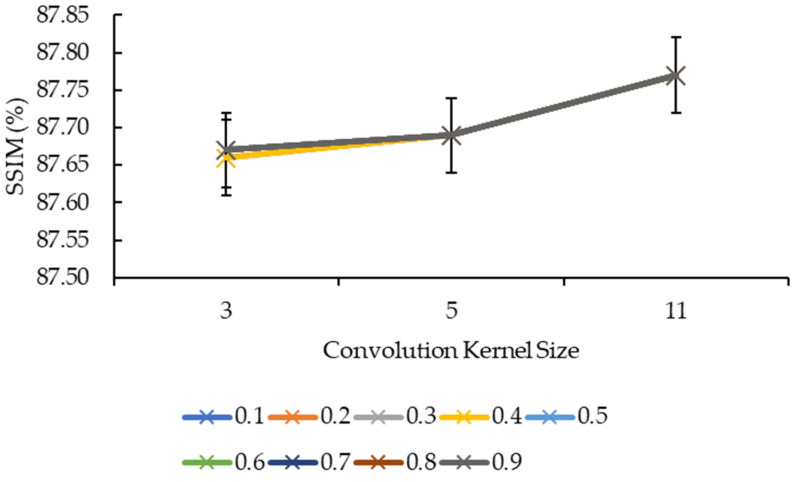
Graph of SSIM against convolution kernel size.

**Figure 4 sensors-21-07955-f004:**
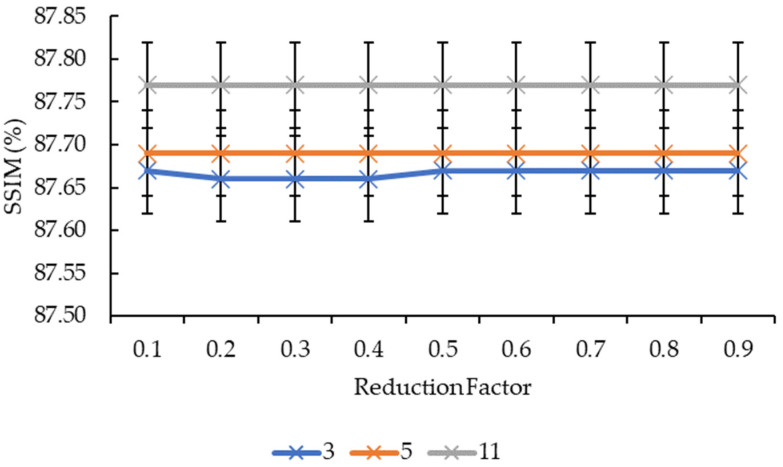
Graph of SSIM against reduction factor.

**Figure 5 sensors-21-07955-f005:**
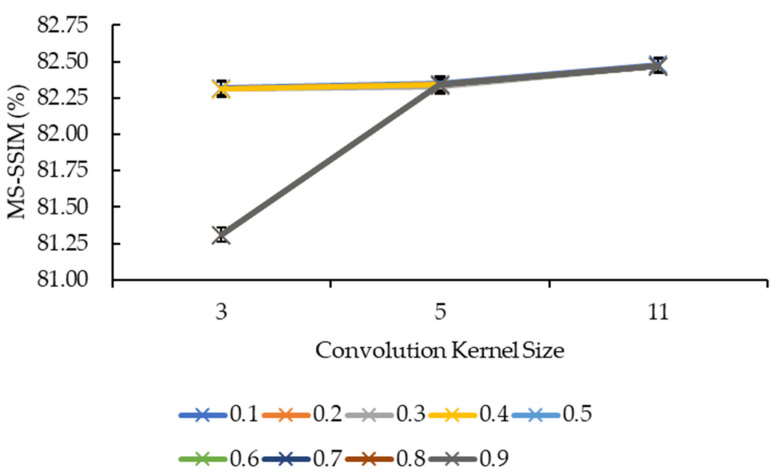
Graph of MS-SSIM against convolution kernel size.

**Figure 6 sensors-21-07955-f006:**
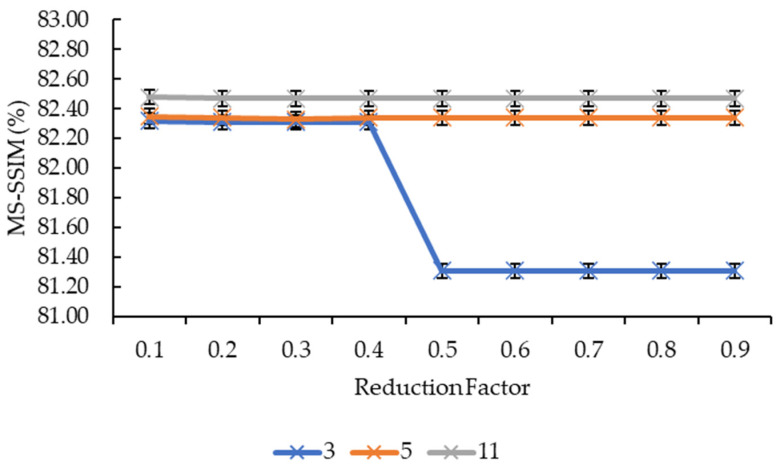
Graph of MS-SSIM against reduction factor.

**Figure 7 sensors-21-07955-f007:**
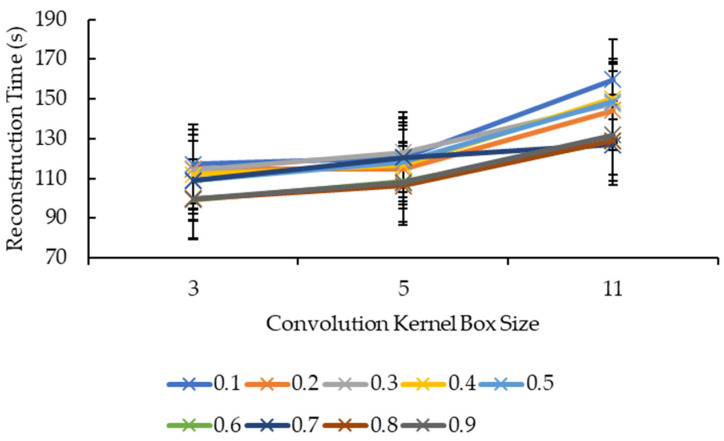
Graph of reconstruction time against convolution kernel size for convolution kernel box.

**Figure 8 sensors-21-07955-f008:**
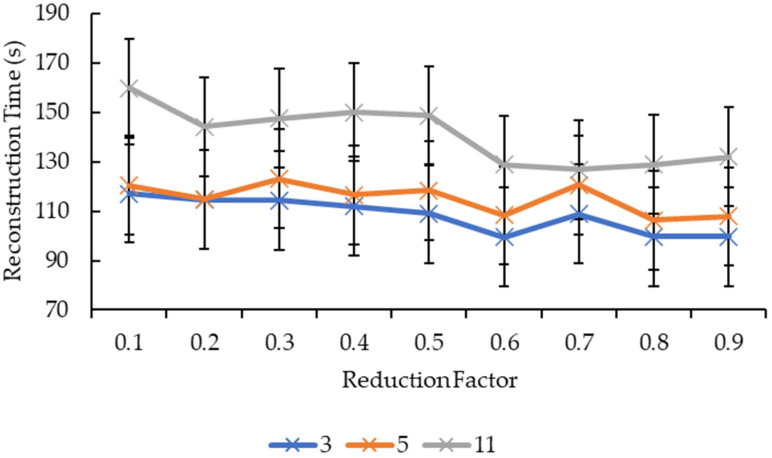
Graph of reconstruction time against reduction factor for convolution kernel box.

**Figure 9 sensors-21-07955-f009:**
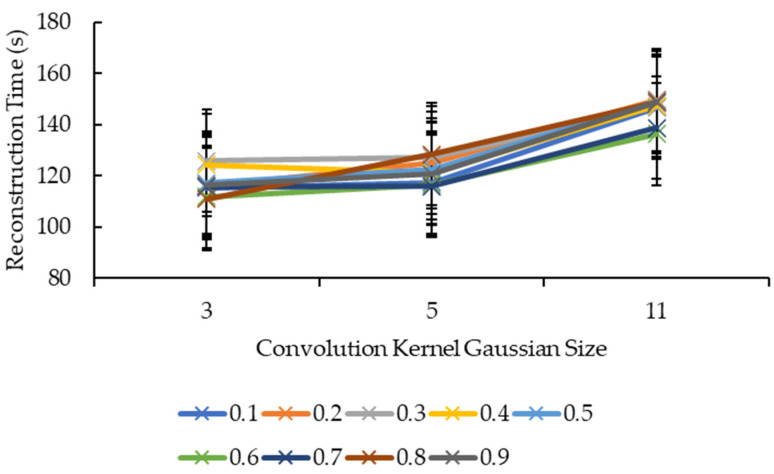
Graph of reconstruction time against convolution kernel size for convolution kernel Gaussian.

**Figure 10 sensors-21-07955-f010:**
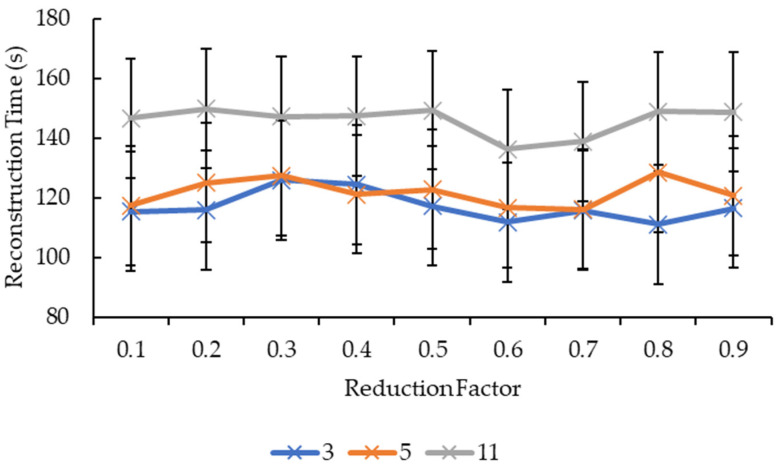
Graph of reconstruction time against reduction factor for convolution kernel Gaussian.

**Figure 11 sensors-21-07955-f011:**
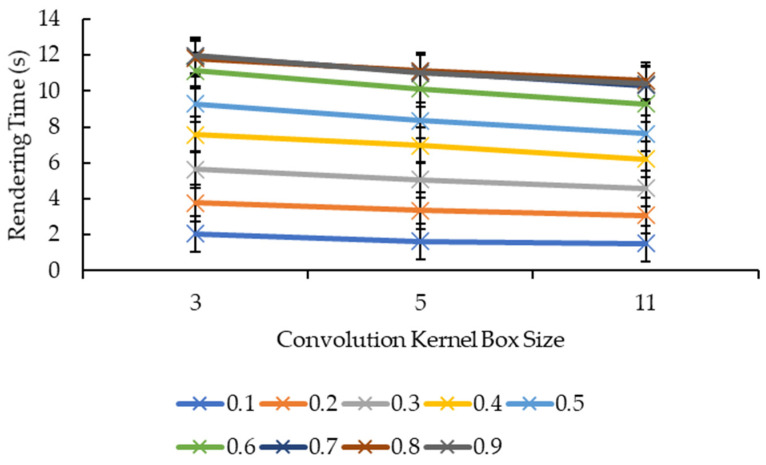
Graph of rendering time against convolution kernel size for convolution kernel box.

**Figure 12 sensors-21-07955-f012:**
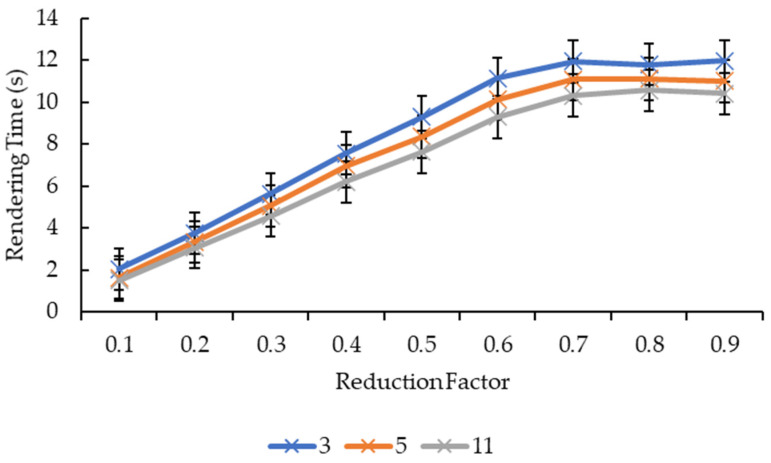
Graph of rendering time against reduction factor for convolution kernel box.

**Figure 13 sensors-21-07955-f013:**
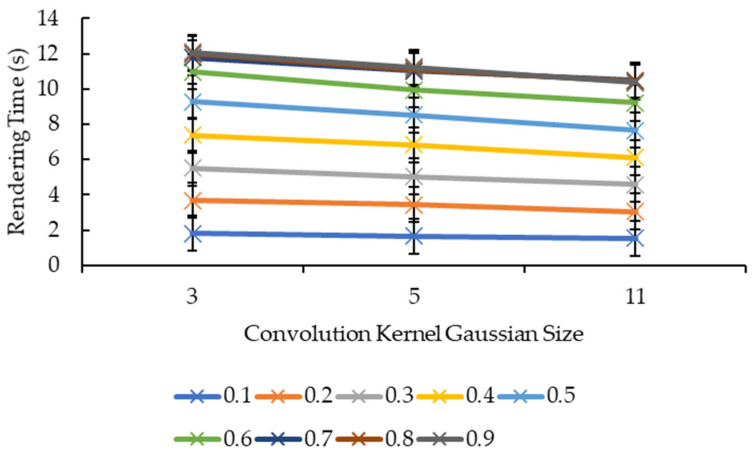
Graph of rendering time against convolution kernel size for convolution kernel Gaussian.

**Figure 14 sensors-21-07955-f014:**
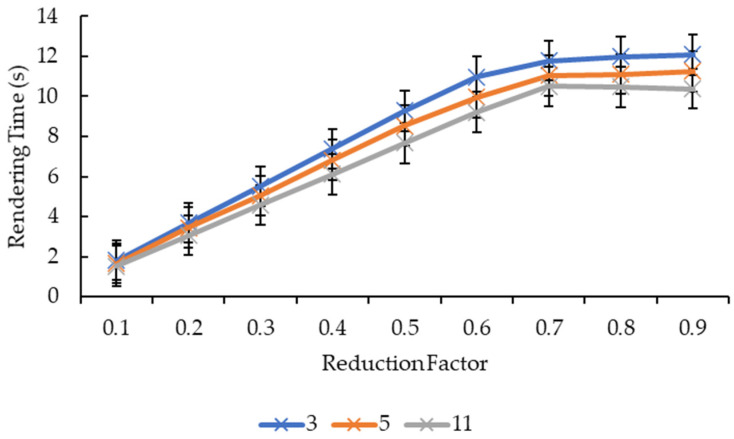
Graph of rendering time against reduction factor for convolution kernel Gaussian.

**Figure 15 sensors-21-07955-f015:**
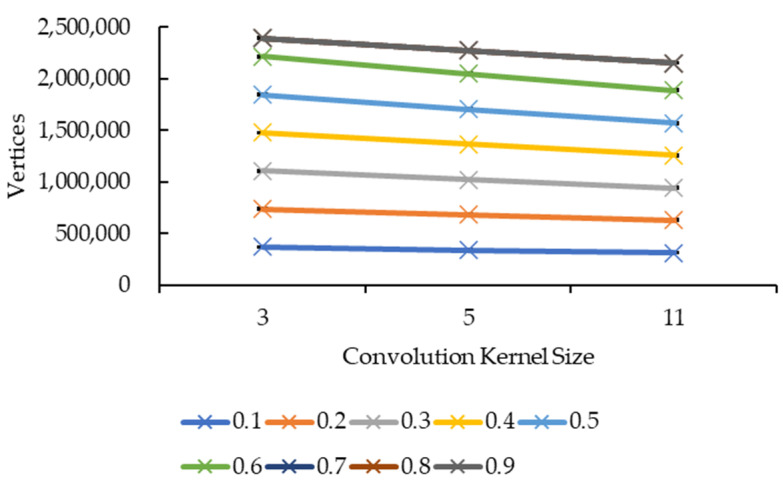
Graph of number of vertices against convolution kernel size.

**Figure 16 sensors-21-07955-f016:**
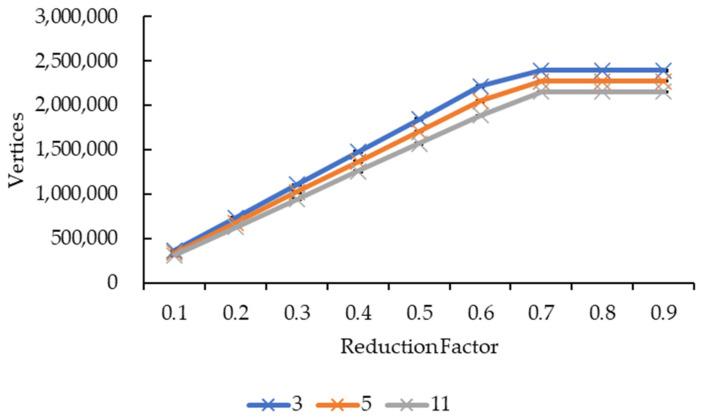
Graph of number of vertices against reduction factor.

**Figure 17 sensors-21-07955-f017:**
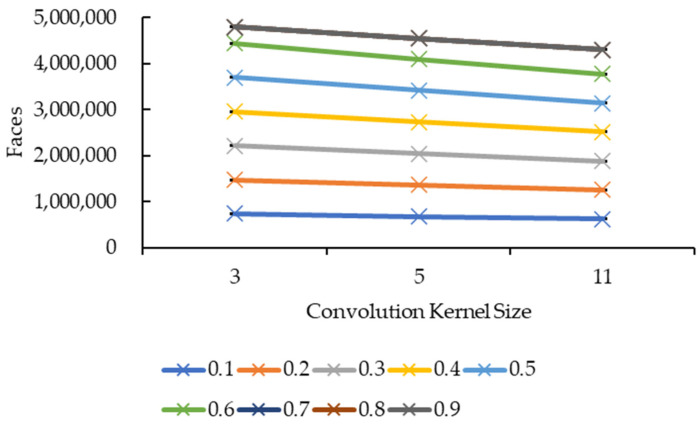
Graph of number of faces against convolution kernel size.

**Figure 18 sensors-21-07955-f018:**
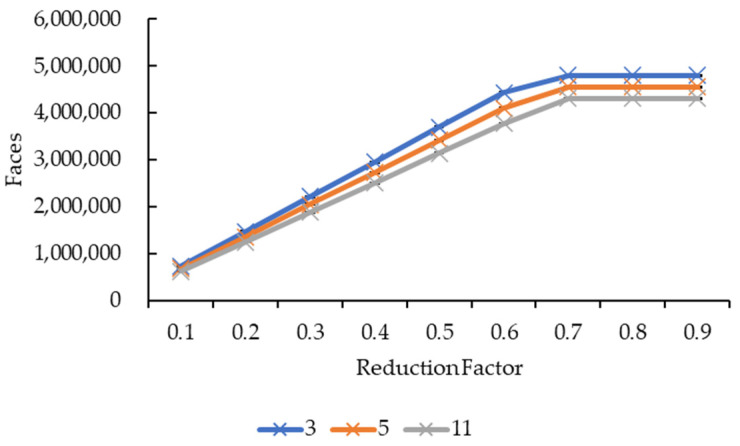
Graph of number of faces against reduction factor.

**Figure 19 sensors-21-07955-f019:**
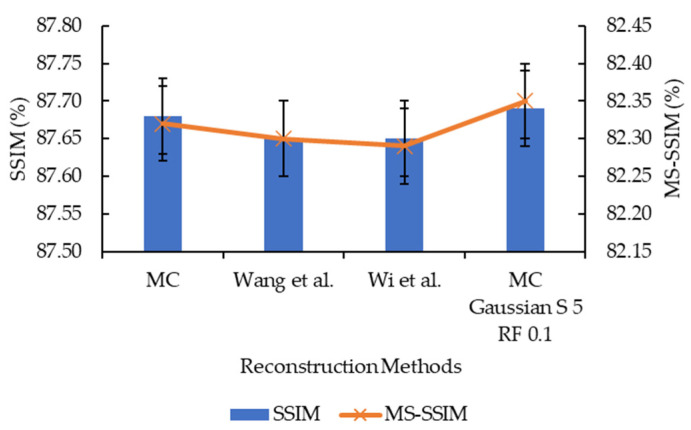
Graph of SSIM and MS-SSIM against different reconstruction methods.

**Figure 20 sensors-21-07955-f020:**
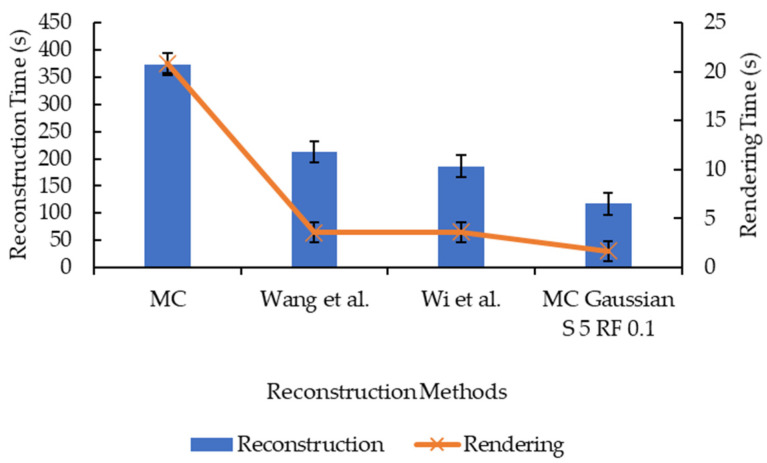
Graph of reconstruction and rendering time against different reconstruction methods.

**Figure 21 sensors-21-07955-f021:**
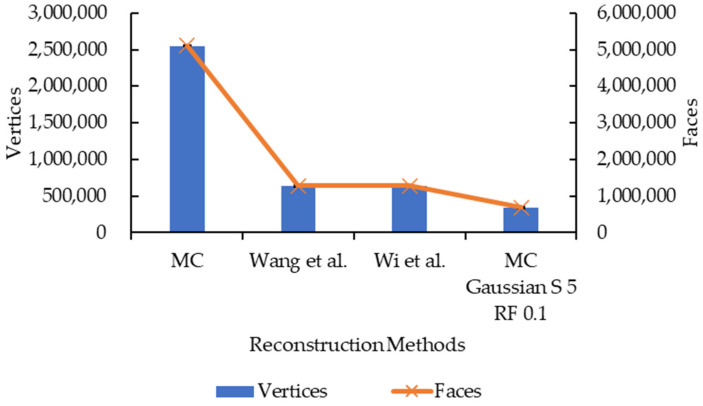
Graph of number of vertices and faces against different reconstruction methods.

**Figure 22 sensors-21-07955-f022:**
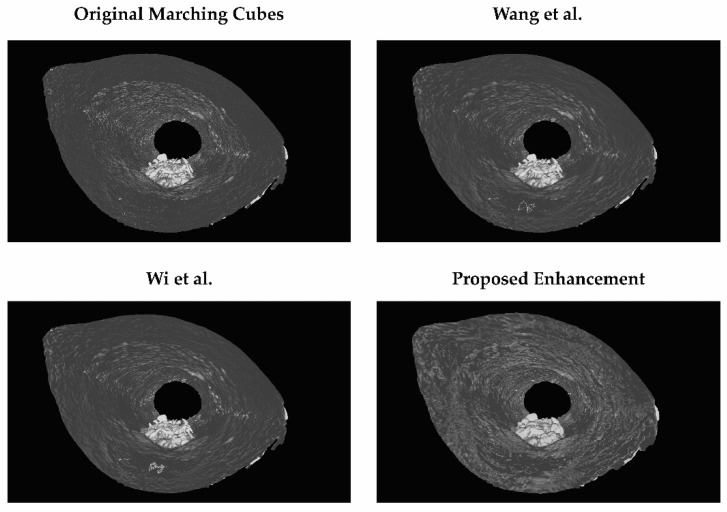
3D models reconstructed with different reconstruction methods.

**Table 1 sensors-21-07955-t001:** SSIM and MS-SSIM for Marching Cubes and Marching Tetrahedra.

Reconstruction Method	SSIM (%)	MS-SSIM (%)
Marching Cubes	87.68	82.32
Marching Tetrahedra	87.66	82.27

## Data Availability

The data presented in this study are available upon request from the corresponding author. The data are not publicly available due to research confidentiality.

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
