# Peer review of "Effects of Different Parameter Settings for 3D Data Smoothing and Mesh Simplification on Near Real-Time 3D Reconstruction of High Resolution Bioceramic Bone Void Filling Medical Images"

_sensors, 2021, doi:10.3390/s21237955_

Round 1
Reviewer 1 Report
Authors investigated different setups in 3D surface reconstruction from 2D CT images and compared the influences in accuracy and efficiency on different parameters. It is rather an exhausted investigation report than a scientific paper.
I am sure this should be very meaningful in this particular area but the authors need to back it up by placing this work among all other works so readers have a better idea of why / how these methods a better than others.
For all figures, only the fitted or connecting lines are illustrated but no data points or error bar are displayed. Please indicate the data points with error bars if available, which will assist a better judgement in the quality of this research.
At lease one image of the reconstructed 3D surfaces with those two methods will be helpful.
Detailed data are good, though Table 2 seems too much for readers. Please try to visualize these data in at least 2 figures (Accuracy with SSIM/MS-SSIM and Reconstruction / Rendering time). This table seems good but can be moved to Appendix as a whole.
Same for Table 3
As indicated in Page 5, this fast algorithm for 3D reconstruction is only available in MATLAB with GPU support. What if the GPU is not NVIDIA or there is no driver support for GPU calculation. Is that only for CUDA?
The authors of this manuscript discussed the influence of different setups on accuracy and execution time. The SSIM and MS-SSIM values are all above 80%. How is this accuracy compare to other methods? Same as the time. Readers may wonder what is the improvement by these algorithms comparing to other works.
Is the whole point of Figure 18 is showing us that there is no difference between different convolution kernels? That is a lot of subfigures to show no differences. If that is the whole purpose of it, maybe two or three images will make the point.
Some expression has to improved, please proof read in revision
For example:
P2:
With the advancement of 3D technologies, 3D reconstruction is seen applied in many 45 areas that require handling tasks using computer vision.
P18
“Based on Figure 2, two possible explanations can be made on the shape of the lines: 449 one, the graph exhibits exponential growth, meaning to say that the increase in SSIM val- 450 ues is proportional to the increase in convolution kernel size.”
On Page 2, the two paragraphs starting with “Surface rendering and volume rendering are common methods applied in visualiz- 56 ing medical images in their equivalent 3D form. B” are lengthy and contains redundant descriptions. Please consider to make them more concise.
When a reference is cited, please mention the name of the main author, not just the number in the reference list, which is not the convention in referencing, as P3
“For example, [13] proposed …”
“called the Marching Cubes 33, is introduced by [15], 94 which is further extended by [16] by grouping the cube configurations into either simple 95 leaves triangulation, tunnel triangulation, or interior point leaves triangulation”
The author of this manuscript never mentioned a name in citations.
Author Response
Authors investigated different setups in 3D surface reconstruction from 2D CT images and compared the influences in accuracy and efficiency on different parameters. It is rather an exhausted investigation report than a scientific paper.
Thank you for the feedback.
I am sure this should be very meaningful in this particular area but the authors need to back it up by placing this work among all other works so readers have a better idea of why / how these methods a better than others.
We agree on this point and have included two works from other researchers to be compared with our work. The results are visualized in three different figures (Accuracy with SSIM, MS-SSIM; Accuracy with reconstruction time, rendering time; Accuracy with number of vertices and faces).
For all figures, only the fitted or connecting lines are illustrated but no data points or error bar are displayed. Please indicate the data points with error bars if available, which will assist a better judgement in the quality of this research.
Thank you for pointing this out. We have made the changes accordingly by including the data points with error bars for all the graphs.
At lease one image of the reconstructed 3D surfaces with those two methods will be helpful.
Thank you for pointing this out. We have included images on the reconstructed 3D surfaces as shown in Figure 22.
Detailed data are good, though Table 2 seems too much for readers. Please try to visualize these data in at least 2 figures (Accuracy with SSIM/MS-SSIM and Reconstruction / Rendering time). This table seems good but can be moved to Appendix as a whole.
We agree with the reviewer's comment and have moved Table 2 to Appendix as a whole.
Same for Table 3
We agree with the reviewer's comment and have moved Table 3 to Appendix as a whole.
As indicated in Page 5, this fast algorithm for 3D reconstruction is only available in MATLAB with GPU support. What if the GPU is not NVIDIA or there is no driver support for GPU calculation. Is that only for CUDA?
Thank you for your question. MATLAB only supports CUDA enabled NVIDIA GPU, hence it will hit an error if any other GPU is used like AMD. If there is no GPU, the program will run without assessing the memory in GPU. The parallel for loop parfor will still be valid as it relies on the CPU cores. This is a potential limitation to the study, hence we have added this as part of the potential improvement as such: For future studies, several things can be considered: 12. Support different GPU products like GPU from AMD.
The authors of this manuscript discussed the influence of different setups on accuracy and execution time. The SSIM and MS-SSIM values are all above 80%. How is this accuracy compare to other methods? Same as the time. Readers may wonder what is the improvement by these algorithms comparing to other works.
We agree with the reviewers' comment and have included two works from other researchers to be compared with our work in terms of accuracy, time, and number of vertices and faces. The comparison is illustrated in three different figures (Accuracy with SSIM, MS-SSIM; Accuracy with reconstruction time, rendering time; Accuracy with number of vertices and faces).
Is the whole point of Figure 18 is showing us that there is no difference between different convolution kernels? That is a lot of subfigures to show no differences. If that is the whole purpose of it, maybe two or three images will make the point.
Thank you for the suggestion. Figure 2 (previously known as Figure 18) shows all the reconstructed surfaces with different parameter values. We believe it is a good idea to show all the reconstructed surfaces in the paper; hence the figure.
Some expression has to improved, please proof read in revision
Thank you for pointing this out. Some of the sentences are rephrased.
With the advancement of 3D technologies, 3D reconstruction is seen applied in many 45 areas that require handling tasks using computer vision.
We agree with the reviewer and have modified this sentence to “With the advancement in 3D technologies, more and more domains are adopting 3D reconstruction technologies.”.
“Based on Figure 2, two possible explanations can be made on the shape of the lines: 449 one, the graph exhibits exponential growth, meaning to say that the increase in SSIM val- 450 ues is proportional to the increase in convolution kernel size.”
We agree with the reviewer and have modified this sentence to “Based on Figure 3, two possible explanations can be made on the shape of the lines: one, the increase in SSIM values is proportional to the increase in convolution kernel size.”.
On Page 2, the two paragraphs starting with “Surface rendering and volume rendering are common methods applied in visualiz- 56 ing medical images in their equivalent 3D form. B” are lengthy and contains redundant descriptions. Please consider to make them more concise.
We agree with the reviewer and have made thorough changes to the two paragraphs.
When a reference is cited, please mention the name of the main author, not just the number in the reference list, which is not the convention in referencing, as P3
“For example, [13] proposed …”
“called the Marching Cubes 33, is introduced by [15], 94 which is further extended by [16] by grouping the cube configurations into either simple 95 leaves triangulation, tunnel triangulation, or interior point leaves triangulation”
The author of this manuscript never mentioned a name in citations.
Thank you for pointing this out. We have made changes so that the main author is mentioned for citations that requires the authors to be named.
Reviewer 2 Report
The explanations in the manuscript are unorganised (introduction and methodology), and inconsistent thus needs improvement.
What are the main objectives? Please clarify since;
In Line 53 it is claimed that "The goal is to compare 3D reconstruction algorithms which are generally grouped into two categories: surface rendering and volume rendering"? The authors argued that the project will focus on surface rendering; please clarify?
Line 62-"many of the improvements focused on increasing the reconstruction accuracy and very few real-time or near real-time surface rendering of large 3D bone defect models"? How has this issue been addressed?
Line 64-". Many of the improved methods that result in reduced reconstruction time are not tested with large image datasets. Not many papers investigated the effects of the different parameter settings of improvement methods towards the reconstruction accuracy, reconstruction time, and optimisation of vertices and faces.
As the main focus is on visualising the surface of the bone defect, the three main objectives in this study are one, to compare different surface rendering techniques in terms of reconstruction accuracy; two, to optimise the better surface rendering technique towards near real-time rendering and higher reconstruction accuracy by experimenting on different parameter value combinations; three, to study the effects of the improvements on the reconstruction accuracy, reconstruction time, rendering time, and the number of vertices and faces. "? The conclusion does not address how the objectives are met!
The authors claim that "The reconstruction speed is subjective to the size of the medical images being reconstructed". It is an obvious claim but has the study tried various image sizes.
Apparently, the authors have implemented the algorithms available in Metlab for their optimisation. Please clarify if it is the case? or "The Marching Cubes code [25], and The Marching Tetrahedra code written in C by [26]"?
In the Marching cube code, what is the cube configurations (15/21/33) in this study? Why?
In the introduction, many improvements on both algorithms were surveyed. Please link the literature to your research? which one has been adopted in this study? And why?
In the methodology section, the authors stated that two steps have been added to the existing Matlab algorithm; "the 3D data smoothing step and mesh simplification step". Is it the main contribution?
In conclusion, the authors claim that "Marching Cubes have higher reconstruction accuracy than Marching line Tetrahedra for large bone defect medical images". Is the decision based on real-time or near-real-time rendering? In the abstract Line 79 it is already mentioned that the Marching Cube algorithm is a generic method; it is quick and straightforward? What is the contribution of this study?
Line 81- It is claimed that "The reconstruction speed is subjective to the size of the medical images being reconstructed". How many images have been
How has the ambiguity issue during the triangulation step been addressed here?
Line 251- "It is noticed that regardless of the convolution kernel, as long as the size and reduction factors are the same, they will have the same SSIM, MS-SSIM, and number of vertices and faces."
Author Response
The explanations in the manuscript are unorganised (introduction and methodology), and inconsistent thus needs improvement.
Thank you for pointing this out. We have rearranged our introduction and methodology. Hopefully the introduction and methodology are improved with the changes.
What are the main objectives? Please clarify since;
Thank you for the comment. We have modified this to: “Hence, this study focuses on three objectives:
- To compare Marching Cubes and Marching Tetrahedra in terms of reconstruction accuracy for large image dataset;
- To optimize the better surface rendering technique towards near real-time ren-dering and higher reconstruction accuracy, lower reconstruction and rendering time, and lower number of vertices and faces by experimenting on different pa-rameter value combinations;
- To study the effects of the improvements’ different parameter values on the re-construction accuracy, reconstruction time, rendering time, and the number of vertices and faces.”.
In Line 53 it is claimed that "The goal is to compare 3D reconstruction algorithms which are generally grouped into two categories: surface rendering and volume rendering"? The authors argued that the project will focus on surface rendering; please clarify?
Thank you for pointing this out. We have included : “Therefore, as the main focus is on visualizing the surface of the bone defect, the study will only be focusing on surface rendering methods.”.
Line 62-"many of the improvements focused on increasing the reconstruction accuracy and very few real-time or near real-time surface rendering of large 3D bone defect models"? How has this issue been addressed?
Thank you for the question. We have modified the sentence as such: “Although many of the improvements focused on increasing the reconstruction accu-racy and reducing the reconstruction time for relatively small image datasets, they are not tested with large image datasets, and not many papers made an in-depth study on the effects of the different parameter settings of improvement methods towards the reconstruction accuracy, reconstruction time, and optimization of vertices and faces.”.
Line 64-". Many of the improved methods that result in reduced reconstruction time are not tested with large image datasets. Not many papers investigated the effects of the different parameter settings of improvement methods towards the reconstruction accuracy, reconstruction time, and optimisation of vertices and faces.
Sorry in advance for the misunderstanding but the reviewer did not mention any concern over this sentence. We assume the concern on this particular sentence is similar to the previous one and have made similar changes.
As the main focus is on visualising the surface of the bone defect, the three main objectives in this study are one, to compare different surface rendering techniques in terms of reconstruction accuracy; two, to optimise the better surface rendering technique towards near real-time rendering and higher reconstruction accuracy by experimenting on different parameter value combinations; three, to study the effects of the improvements on the reconstruction accuracy, reconstruction time, rendering time, and the number of vertices and faces. "? The conclusion does not address how the objectives are met!
Thank you for pointing this out. We have modified our conclusion as such: “In conclusion, Marching Cubes have higher reconstruction accuracy than March-ing Tetrahedra for large bone defect medical images. The proposed improvement, which involves 3D data smoothing with convolution kernel Gaussian size 5 and mesh simplification with a reduction factor of 0.1, is the most optimal parameter value com-bination in achieving a balance between high reconstruction accuracy, low reconstruc-tion and rendering time, and a low number of vertices and faces. With help from GPU, parallel processing and vectorization, the proposed enhancement has the potential in the near real-time reconstruction of large bone defect medical images.”.
The authors claim that "The reconstruction speed is subjective to the size of the medical images being reconstructed". It is an obvious claim but has the study tried various image sizes.
Thank you for the question. We have tried various image sizes, but ultimately decided to only include one image dataset. As the study focuses on only large image datasets consisting of high resolution bioceramic bone void filling medical images that can be processed all in one shot without dividing it into batches, amongst all the large image datasets that we have on hand, the one mentioned in the study is the smallest one and the one and only dataset that we can process in one shot. We have removed the claim. Hopefully this answers the reviewer's concern.
Apparently, the authors have implemented the algorithms available in Metlab for their optimisation. Please clarify if it is the case? or "The Marching Cubes code [25], and The Marching Tetrahedra code written in C by [26]"?
Thank you for pointing this out. We have included the main author for the Marching Cubes code and the Marching Tetrahedra code written in C. We have also stated that the functions used are MATLAB built-in functions as such: “The 3D data smoothing step applied in this paper is a MATLAB built-in function called smooth3”.
In the Marching cube code, what is the cube configurations (15/21/33) in this study? Why?
Thank you for the question. All the cube configurations during triangulation step can be generally categorized into 15, 21, and 33 unique cube configurations. 21 and 33 cube configurations are introduced to overcome complex cube topologies. We have added this explanation.
In the introduction, many improvements on both algorithms were surveyed. Please link the literature to your research? which one has been adopted in this study? And why?
Thank you for pointing this out. We have rearranged the introduction section so that the literatures are mentioned first. Also, we have specifically stated the works that we have adopted and replicated in this study for comparison purposes along with its explanation as such: “Besides comparing the original Marching Cubes and the proposed enhancement, works by Wang et al. [13] and Wi et al. [17] are replicated and compared in this study. Their works were selected to replicate as both involve the smoothing step and mesh decima-tion step in their improvement, and the proposed enhancement also involves smooth-ing and mesh decimation; hence a direct comparison can be made.”.
In the methodology section, the authors stated that two steps have been added to the existing Matlab algorithm; "the 3D data smoothing step and mesh simplification step". Is it the main contribution?
Thank you for the question. The two additional steps are the main contribution in this study.
In conclusion, the authors claim that "Marching Cubes have higher reconstruction accuracy than Marching line Tetrahedra for large bone defect medical images". Is the decision based on real-time or near-real-time rendering?
Thank you for the question. We have modified the first objective so that it is reflected in the conclusion as well. The first objective is modified to “To compare Marching Cubes and Marching Tetrahedra in terms of reconstruction accuracy for large image dataset”.
In the abstract Line 79 it is already mentioned that the Marching Cube algorithm is a generic method; it is quick and straightforward? What is the contribution of this study?
Thank you for the question. We have modified the sentence to “The Marching Cubes algorithm is a popular surface rendering method applied in the medical field, mainly because of its implementation simplicity and relatively fast re-construction speed”. The main contribution of this study is the addition of the smoothing and mesh decimation step which results in increased reconstruction accuracy, decreased reconstruction and rendering time, and decreased vertices and faces for large medical image dataset.
Line 81- It is claimed that "The reconstruction speed is subjective to the size of the medical images being reconstructed". How many images have been
Thank you for the question. Although the question is cut off halfway, we believe the question is addressed to how many images have we tested to support the claim. The answer to this is about five image datasets, but amongst them all, only one image dataset that qualifies in this study. We have removed this claim. Hopefully this answers the question.
How has the ambiguity issue during the triangulation step been addressed here?
Thank you for the question. The ambiguity issue is addressed through the 3D data smoothing step. This statement is added in “This step removes noises in the 3D data, which may improve the 3D reconstruction accuracy and solve the ambiguity issue because noises often lead to wrong triangula-tions.”.
Line 251- "It is noticed that regardless of the convolution kernel, as long as the size and reduction factors are the same, they will have the same SSIM, MS-SSIM, and number of vertices and faces."
Sorry in advance for the misunderstanding but we are not sure what concern the reviewer has on this statement.